# An Italian Real-World Study Highlights the Importance of Some Clinicopathological Characteristics Useful in Identifying Metastatic Breast Cancer Patients Resistant to CDK4/6 Inhibitors and Hormone Therapy

**DOI:** 10.3390/biomedicines12030498

**Published:** 2024-02-22

**Authors:** Roberta Maltoni, Andrea Roncadori, William Balzi, Massimiliano Mazza, Fabio Nicolini, Michela Palleschi, Paola Ulivi, Sara Bravaccini

**Affiliations:** IRCCS Istituto Romagnolo per lo Studio dei Tumori (IRST) “Dino Amadori”, 47014 Meldola, Italy; roberta.maltoni@irst.emr.it (R.M.); massimiliano.mazza@irst.emr.it (M.M.); fabio.nicolini@irst.emr.it (F.N.); michela.palleschi@irst.emr.it (M.P.); paola.ulivi@irst.emr.it (P.U.); sara.bravaccini@irst.emr.it (S.B.)

**Keywords:** breast cancer, CDK4/6 inhibitors, hormone therapy, resistance, neutropenia, multistate model, real world

## Abstract

**Background:** Cyclin-dependent kinase 4 and 6 (CDK4 and CDK6) inhibitors have changed the therapeutic management of hormone receptor-positive (HR+) metastatic breast cancer (mBC) by targeting the cell cycle machinery and overcoming endocrine resistance. However, a large number of patients present disease progression due to cancer cells resisting CDK4/6 inhibitors. Our research considers which clinicopathological characteristics could be useful in identifying patients who might respond to CDK4/6 inhibitors by analyzing a retrospective case series of patients with HR+ mBC who were treated with hormone therapy plus CDK4/6 inhibitors. **Methods:** Approximately 177 mBC patients were enrolled, of whom 66 were treated with CD4/6 inhibitors plus letrozole and 111 were treated with CDK4/6 inhibitors and fulvestrant. A multistate model was used. **Results:** A low body surface area and older age were associated with an increased risk of developing neutropenia. A high Ki67 index, the presence of visceral metastases, and not having previously undergone adjuvant chemotherapy were prognostic factors of disease progression/death. As expected, some of the neutropenic patients who had previously undergone multiple lines of treatment were at a higher risk of disease progression/death. Furthermore, neutropenia status was associated with a more than doubled risk of progression/death compared to patients without neutropenia (HR = 2.311; *p* = 0.025). **Conclusions:** Having identified certain factors that could be associated with the development of neutropenia and considering that neutropenia itself is associated with an increased risk of progression, we believe that the baseline characteristics should be taken into account to reduce cases of neutropenia and disease progression.

## 1. Introduction

In 2023, the number of new breast cancer (BC) cases in the US was estimated at 297,790, with a forecasted number of deaths of 43,170 [1]. Estrogen receptor (ER)-positive/c-erb-B2 negative is the most common subtype of BC given that more than 70% of BC patients have ER+ tumors. The ER pathway plays a very important role in the development and progression of BC [2]. Hormonal therapies represent the gold standard treatment of hormone receptor (HR)-positive BC patients, and their use leads to better patient prognosis in both the first- and second-line settings [3,4,5]. The efficacy of hormone therapy is limited due to hormone resistance mechanisms and is an obstacle that clinicians and researchers must overcome [2]. While most ER+ BC patients initially respond to hormone treatment, 15–20% of tumors are intrinsically resistant to treatment, and an additional 30–40% later become treatment resistant. ER-positive BC patients with resistance to endocrine therapy are associated with high rates of mutations and a selection of resistant subclones and are at greater risk of death or relapse [6,7]. Cyclin-dependent kinase (CDK) 4/6 inhibitors have been shown to be effective in the treatment of metastatic HR-positive BC and human epidermal growth factor receptor 2 (HER2)-negative BC combined with hormonal therapy (HT) [8]. In clinical settings, CDK 4/6 inhibitors have changed the therapeutic management of HR+ BC patients by targeting the cell cycle machinery and overcoming aspects of endocrine resistance. Palbociclib, ribociclib, and abemaciclib have been approved in combination with an aromatase inhibitor or fulvestrant for HR+ mBC. Abemaciclib has also been approved as a monotherapy for pre-treated patients [9]. In general, the toxicity profiles of CDK 4/6 inhibitors are similar, but each drug has toxicity specifications. The most common side effects are neutropenia, leukopenia, fatigue, nausea, infections, arthralgia, anemia, headache, and diarrhea. Apart from neutropenia and leukopenia, some patients have reported grade 1 or 2 sensitivity. Cases of grade 3 and 4 neutropenia and leukopenia, which are commonly reported after treatment with palbociclib and ribociclib, are generally managed with dose interruption or reduction. Cases of interstitial pneumonia are uncommon but of notable clinical entity. Abemaciclib is associated with a significantly higher incidence of diarrhea and increased creatinine. Abemaciclib has recently been approved as an adjuvant treatment in women undergoing ER+/HER-negative BC with a high risk of relapse (positive lymph nodes, tumor > pT2, and elevated Ki67). However, most patients will eventually present disease progression, suggesting that cancer cells resist CDK4/6 inhibitors. Various potential cell cycle-specific and non-specific mechanisms of resistance to CDK4/6 inhibitors have been described in the literature [10,11]. Several retrospective studies and randomized trials have reported that 5–10% of patients develop acquired *RB1* mutation as a mechanism of resistance to CDK4/6 inhibitors, but these are less common due to intrinsic drug resistance. Increased activity of the CDK4/6 checkpoint kinase, bypassing the checkpoint through activation of CCNE1/CDK2, leading to downstream phosphorylation of retinoblastoma (RB) protein or acquired *RB1* loss of function mutations have been described as multiple mechanisms of resistance to CDK4/6 inhibitors [12,13,14,15,16,17]. Other mechanisms, such as upregulation of cyclin E, overexpression of CDK 7, and dysregulation of several signaling pathways such as the Phosphatidylinositol 3-kinase-AKT-mammalian target of rapamycin (*PI3/AKT/mTOR*) pathway, have been described as the main causes of resistance to CDK4/6 inhibitors [18]. HT resistance is often caused by estrogen receptor 1 (*ESR1*) driver mutations in the alpha subunit of the ER or endocrine-independent activity in the *PI3K/AKT/mTOR* pathway [19,20,21]. Up to 40% of patients develop an *ESR1* mutation during treatment with CDK4/6 inhibitors plus aromatase inhibitors, reflecting endocrine resistance [22]; these patients may have continued sensitivity to CDK4/6 inhibition, providing a rationale for maintaining CDK4/6 inhibitors beyond progression and targeting *ESR1* by changing the endocrine therapy to an estrogen receptor degrader or downregulator (SERD). Despite these findings on the role of new emerging biomarkers, we wondered if clinicopathological characteristics could be useful in identifying patients who might respond to CDK4/6 inhibitors by analyzing a retrospective case series of HR-positive, HER2-negative mBC patients receiving HT plus CDK4/6 inhibitors [23]. We conducted the study at our institute (IRCCS Istituto Romagnolo per lo Studio dei Tumori (IRST) “Dino Amadori”).

## 2. Materials and Methods

### 2.1. Study Design

To understand if clinicopathological characteristics could be useful in identifying HR-positive BC patients responsive to HT plus CDK4/6 inhibitors, we reviewed a case series of advanced BC patients in a real-world experience.

### 2.2. Data Source and Patient Selection

Data were retrieved through record-linkage processes between multiple databases, forming part of the institution’s electronic medical record (EHM). All data mining, transformation, and construction processes of the final data were internally validated and checked by informatic specialists. The data underwent further checks carried out by a medical oncologist to ensure completeness and quality. Corrections were made where necessary. A medical review check was performed to check the quality and consistency of the data.

Our retrospective study was performed in compliance with the 1964 Declaration of Helsinki guidelines and was approved by the Medical Committees of the Area Vasta Romagna, Italy (approval number 3692).

#### 2.2.1. Inclusion Criteria

Between 5 July 2017, and 13 April 2021, we enrolled females with the following criteria: HR-positive BC, HER2-negative mBC > 18 years old, who had received first- or second-line treatment and were subsequently treated with HT and CDK4/6 inhibitors (palbociclib, ribociclib, and abemaciclib).

#### 2.2.2. Exclusion Criteria

Patients with DCIS or non-invasive BC were excluded from the study, as well as patients with mBC not treated with HT and CDK4/6 inhibitors.

## 3. Statistical Methods

This is an observational, monocentric, retrospective, secondary data use study. Due to the non-randomized nature of this study, a formal sample size statistical calculation was not performed. Descriptive statistics for the demographic and clinical baseline characteristics were reported. The mean and standard deviation were reported for all continuous variables when the normality distribution assumption was confirmed; otherwise, the median and interquartile ranges were reported. Categorical variables were summarized through counts and percentages. Kaplan–Meier curves were drawn for progression-free survival (PFS). Cox proportional hazard models were developed to estimate hazard ratios: both univariate and multivariable models were fit. The proportional hazards assumption was tested for each covariate in the models; furthermore, a global test was run. A time pattern was not observed when plotting the scaled Schoenfeld residuals against the transformed time curve. Lastly, a good parallelism was observed for the log–log survival curves. To assess the effect of neutropenia regarding the risk of progression/death, multistate models were developed to analyze transitions between the treatment state (event-free), neutropenia state, and progression/death (Figure 1) [24].

Specifically, multistate models analyze how individuals or things occupy and transition between different states to reach a final endpoint. Clinically, the significant baseline characteristics in each transition were modeled using the Cox proportional hazards model. The last set of variables included in the multivariable model was selected based on the Akaike information criterion (AIC). Transition probabilities and state occupation probabilities were calculated using the Aalen–Johansen estimator. Plots were drawn to show any relevant factors affecting the transition probabilities among states, namely the presence of visceral metastases, CDK 4/6 inhibitor lines of therapy, prior adjuvant chemotherapy, patient age, Ki67 expression levels, and body surface area (BSA). The *p*-values were two-sided and the confidence intervals were at 95%. Statistical analyses were performed using R statistical software, version 4.2.0 (www.r-project.org); the mstate R package was used to develop the multistate models.

## 4. Results

### 4.1. Baseline Characteristics

A total of 177 metastatic BC patients were enrolled in the study, of whom 66 (37.3%) were treated with CDK4/6 inhibitors plus letrozole and 111 (62.7%) were treated with CDK4/6 inhibitors and fulvestrant. As reported in Appendix A, out of the 66 patients who underwent CDK4/6 inhibitors plus letrozole treatment, 21 (31.8%) had received prior adjuvant treatment based on chemotherapy followed by hormonal therapy. Out of the 111 patients who were treated with CDK4/6 inhibitors plus fulvestrant, a large percentage had undergone previous adjuvant chemotherapy (N = 50, 45.0%). The median age was 63 years (range 38–92), of whom 103 (58.2%) < 65 years and 74 (41.8%) > 65 years. Approximately 152 (85.9%) patients were postmenopausal and 25 (14.1%) were premenopausal.

### 4.2. Role of Clinicopathological Characteristics concerning the Progression

A total of 86 (49.2%) patients had a high body mass index (BMI) (>25 kg/m^2^), while 90 (50.8%) had a low BMI (Appendix A). Almost the entire patient cohort (N = 175, 98.9%) was ER-positive, and a similar proportion of patients (N = 139, 78.5%) were PgR-positive. Notably, two patients who were ER-negative were PgR-positive. Ki67 was low (<20%) in 85 (48.0%) patients and high (>20) in 92 (52.0%) patients. Approximately 45 (25.4%) patients had bone metastases, 67 patients had visceral metastasis (37.9%), and 65 (36.7%) patients had metastases in other organs. In particular, 75 (42.4%) patients had only one organ involved, 73 (41.2%) presented metastasis in two organs, and the remaining 29 (16.4%) patients had at least three organs involved. The remaining clinical pathological characteristics and previous endocrine and chemotherapy treatments are shown in Table 1.

The PFS analysis was based on two BMI groups (≤25 vs. >25 kg/m^2^), and no statistically significant differences were observed (Appendix A).

After analyzing the factors that could potentially increase the risk of progression/death (results from the multivariable Cox PH model), having received previous adjuvant treatment and the number of metastases were the only two variables that were statistically significant (Appendix A). Specifically, patients undergoing a second (or subsequent) metastatic treatment line were associated with a hazard ratio of 1.78 (95% CI: 1.22–2.59). Similarly, the presence of at least three metastases was associated with an increased risk of progression (hazard ratio: 2.10; 95% CI: 1.35–3.28).

Regarding the clinical characteristics analyzed, neutropenia was absent in 98 (55.4%) patients and present in 79 patients (Appendix A). To deepen our understanding of the mechanisms linking PFS and neutropenia, a multistate model (with neutropenia as a transition state) was developed. High BSA (Figure 2A and Table 2) was associated with a lower risk of developing neutropenia (hazard ratio: 0.026; *p* ≤ 0.0001), whereas older age (>65 years) (Figure 2B) was a prognostic factor of neutropenia (hazard ratio: 1.023; *p* = 0.023). Some of the parameters that were used in the model to predict the risk of death/disease progression after CDK4/6 inhibitor treatment were high Ki67 (Figure 2C), the presence of visceral metastases (Figure 2D), and the absence of previous adjuvant chemotherapy (Figure 2E), which showed a statistically significant association with an increased risk of progression/death (Table 2). As expected, among neutropenic patients, those who underwent multiple previous lines of treatment were at higher risk of disease progression/death (Figure 2F). Finally, as reported in Table 2, having passed through the neutropenia state was associated with a more than doubled risk of progression/death compared to patients without neutropenia (hazard ratio: 2.311; *p* = 0.025).

Analyzing the risk factors using the multivariable Cox PH model showed that having received a previous line of treatment (≥2) and the presence of >2 metastasis were associated with a higher risk of progression (HR 1.224–2.588 *p* = 0.003; HR 1.353–3.277 *p* = 0.001 respectively) (Appendix A).

## 5. Discussion

Many studies have shown that adding CDK4/6 inhibitors to hormone therapy significantly improves the progression-free survival of patients with endocrine-sensitive or endocrine-resistant BC [14,25]. The gold standard treatment for HR-positive metastatic BC patients (stage IIIB-IV) is HT plus CDK4/6 inhibitors. Clinical trials such as the PALOMA-3 trial have shown that CDK4/6 inhibitors with endocrine therapy (ET) have improved overall survival [26]. The FDA-approved CDK4/6 inhibitors used in this clinical setting are palbociclib, ribociclib, and abemaciclib [25]. Several trials and results have led to studies on the use of CDK4/6 inhibitors, along with the FDA approval expansions. Although the use of CDK4/6 inhibitors has improved PFS in patients with hormone receptor-positive, HER2−breast cancer, studies have shown that resistance pathways can lead cells to be insensitive to CDK4/6 inhibitors, leading to continued cell proliferation [27,28]. The percentage of patients who are unresponsive or refractory to these therapies is high, and no reliable and reproducible biomarkers have been validated to select a priori responders or refractory patients [25,29]. Gao et al. conducted predefined subgroup analyses on patients with progesterone receptor disease, a disease-free interval of 12 months or less, de novo metastases, lobular histology, and bone-only disease, visceral metastases, and those aged up to 40 years, and all cases had a clinical benefit from CDK4/6i-endocrine combo therapy [29]. Recently, Cordani and colleagues generated a human MCF-7 luminal breast cancer cell line that was able to survive and proliferate at different palbociclib concentrations and that also showed cross-resistance to abemaciclib. Among the top deregulated genes, they found dramatic downregulation of the CDK4 inhibitor CDKN2B and upregulation of the TWIST1 transcription factor. They concluded that TWIST1 upregulation is a potential target for reversing resistance to the CDK4/6 inhibitor in metastatic luminal breast cancer cells [30]. However, the mechanisms of resistance need to be further investigated both in luminal and ductal breast cancer.

Shikanai A and colleagues analyzed the clinicopathological features related to the efficacy of CDK4/6 inhibitor-based treatments in metastatic breast cancer in Japan [31]. They reported that progression-free survival (PFS) was significantly shorter in patients whose primary tumor was high grade (*p* = 0.016) or in those with a high neutrophil-to-lymphocyte ratio (NLR) at baseline (*p* = 0.017). Meanwhile, in concordance with what we observed in our previous study [32], there were no differences in other factors such as the expression levels of hormone receptors. The authors observed that patients whose metastatic lesions were of low tumor grade or high Ki67 index had longer PFS, and such trends were more obvious than primary lesions. Despite their data suggesting that tumor grading in primary lesions and NLR are potential predictive factors for CDKi-based treatments, pathological assessment of metastatic lesions might also be necessary.

The efficacy of adjuvant treatment in patients of monarchE subpopulation Cohort 1 with high-risk early BC in Japan and Europe has recently been published. The authors analyzed the impact of clinicopathological features that can easily be identified as part of routine clinical BC evaluation on patients’ responses. Efficacy data from Cohort 1 demonstrate substantial evidence of benefit for adjuvant abemaciclib+endocrine therapy in patients with HR+, HER2− early BC at high risk of recurrence (ClinicalTrials.gov: NCT03155997 [monarchE]). However, these results were obtained from patients with early BC [33].

Given the small number of patients analyzed, we are aware of the study’s limitations, and therefore, we did not perform a sub-analysis of the clinical–pathological parameters according to the different CDK4/6 treatments.

One of our study’s strong points that differs from those cited in this text is that we assessed patients’ typical clinicopathological characteristics and several tumors to see if they could have an impact on CDK4/6 inhibitor responses in mBC patients. At our cancer institute, we analyzed patient age, tumor grading, HR status, Ki67, BMI, BSA, previous treatment with adjuvant therapy (chemo or hormone therapy), the number of treatment lines, neutropenia, the number of metastatic sites, and which organs were involved. We found that only having received previous adjuvant treatment and the number of metastases were the only factors associated with PFS. Introducing neutropenia into the Cox PH model resulted in it being a potential predictor of progression.

Lastly, we observed that a low body surface area and older age (>65 years) were associated with an increased risk of developing neutropenia.

### Clinical and Practical Aspects of the Study

The results could have a clinical impact given that all mBC patients received first-line treatment with endocrine therapy plus CD4/6 inhibitors and each patient was given the same oral dose regardless of age, weight, BMI, and BSA. According to our findings, this signifies that a patient with a low BMI can develop a severe neutropenia level, even if the literature data are discordant [34]. Interestingly, our results showed that grading and BMI do not seem to have an impact on PFS.

Having identified some of the characteristics associated with a higher risk of progression, we would like to underline that baseline variables must be taken into consideration when choosing a course of treatment to limit the insurgence of neutropenia and disease progression. In cases where patients have a higher risk of disease progression (i.e., high Ki67 and visceral crisis), clinicians should consider treating patients with chemotherapy. The data need to be validated and improved by increasing patient enrollment and through a long-term follow-up program.

## 6. Conclusions

Our findings confirm that a low body surface area and older age (>65 years) are associated with an increased risk of developing neutropenia, results which we believe should be taken into account in clinical practice before starting first-line therapy with HT and CDK4/6 inhibitors given that they have been associated with a more than doubled risk of progression/death compared to non-neutropenic patients. Indeed, neutropenic patients who have undergone multiple treatment lines should be followed up closely since they are at a higher risk of disease progression/death. Lastly, high Ki67, the presence of visceral metastases, and the absence of prior adjuvant chemotherapy should also be considered in clinical practice given that they could be prognostic factors of progression/death.

## Figures and Tables

**Figure 1 biomedicines-12-00498-f001:**
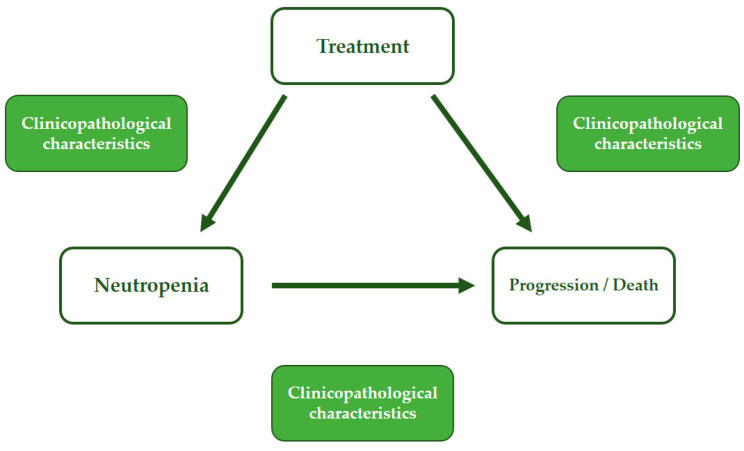
Multistate model structure.

**Figure 2 biomedicines-12-00498-f002:**
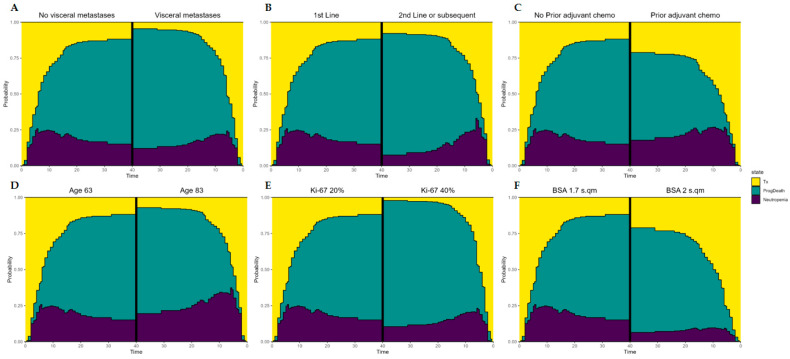
State occupancy probability plot based on (**A**) the presence of visceral metastases, (**B**) having received first-line or second-line or subsequent lines of therapy, (**C**) having previously received adjuvant chemotherapy, (**D**) patient age, (**E**) Ki67 expression, and (**F**) body surface area (BSA).

**Table 1 biomedicines-12-00498-t001:** Baseline characteristics of patients in the overall population and stratified by treatment.

Baseline Characteristics	(n = 177)
**Age**	
Median (min/max)	63 (38–92)
≤65 y	103 (58.2%)
>65 y	74 (41.8%)
**Menopausal status**	
Postmenopause	152 (85.9%)
Pre/perimenopause	25 (14.1%)
**BMI**	
≤25	90 (50.8%)
>25	87 (49.2%)
**Previous hormonal therapy (HT)**	
No	42 (23.7%)
Yes	135 (76.3%)
**Prior aromatase inhibitor (from the total number of patients receiving HT)**	
No	33 (24.4%)
Yes	102 (75.6%)
**Prior adjuvant chemotherapy**	
No	106 (59.9%)
Yes	71 (40.1%)
**ER status (positive if the value ≥ 10%)**	
Positive	175 (98.9%)
Negative	2 (1.1%)
**PgR status (positive if the value ≥ 10%)**	
Positive	139 (78.5%)
Negative	38 (21.5%)
**Previous lines**	
First line	95 (53.7%)
≥2 lines	82 (46.3%)
**Metastatic site**	
Bone only	45 (25.4%)
Visceral	67 (37.9%)
Other	65 (36.7%)
**Organs involved, N°**	
1	75 (42.4%)
2	73 (41.2%)
≥3	29 (16.4%)
**Neutropenia**	
Absent	79 (44.6%)
Present	98 (55.4%)
**Ki67 (%)**	
Median (min/max)	20 (0–80)
<20%	85 (48.0%)
≥20%	92 (52.0%)
**HT**	
Fulvestrant	111 (62.7%)
Letrozole	66 (37.3%)
**Grading**	
1	3 (1.9%)
2	98 (62.8%)
3	55 (35.3%)
Missing/not known	21

**Table 2 biomedicines-12-00498-t002:** State transition Cox multistate hazard model.

	Parameter	Coefficient	CI-Low	CI-High	*p*-Value
**Treatment >** **Neutropenia**	Previous lines (2L+)	1.455	0.933	2.269	0.0982
	Baseline BSA	0.026	0.005	0.123	<0.0001
	Age	1.023	1.003	1.043	0.0229
**Treatment >** **progression/death**	Ki67 (%)	1.040	1.022	1.058	<0.0001
	Visceral metastases	1.680	1.022	2.762	0.0408
	Prior adjuvant chemotherapy	0.565	0.335	0.954	0.0326
**Neutropenia >** **progression/death**	Previous lines (2L+)	1.893	1.059	3.384	0.0312
	Neutropenia	2.311	1.109	4.815	0.0253

## Data Availability

All data are in the manuscript and the Appendix A.

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
