# Peer review of "An Italian Real-World Study Highlights the Importance of Some Clinicopathological Characteristics Useful in Identifying Metastatic Breast Cancer Patients Resistant to CDK4/6 Inhibitors and Hormone Therapy"

_biomedicines, 2024, doi:10.3390/biomedicines12030498_

Round 1

Reviewer 1 Report

Comments and Suggestions for Authors

First of all, I would like to thank you for inviting me to review the manuscript entitled: 'Which are the clinicopathological characteristics useful to define the metastatic breast cancer patients that will respond to CDK4/6 inhibitors and hormone therapy?: an Italian real world experience’. The aim of the study was to estimate the clinicopathological characteristics could be useful to identify the patients that will respond to CDK4/6 inhibitors by the analysis of a retrospective case series of patients with HR+ mBC treated with hormone therapy plus CDK4/6 inhibitors. The general conclusion demonstrated that identified a set of factors associated with the probability to develop neutropenia and considering that neutropenia itself is associated with an increased risk of progression, these baseline characteristics should be taken into account in order to reduce the occurrence of both neutropenia and disease progression. Thus, I recommend publication after some major issues have been addressed:

Major:
1. In Methods section, authors should provide subsections, including study design, inclusion and exclusion criteria, etc.
2. In my version of the pdf file, figure 1 is incomplete. Figure 2 is also incomplete.
3. In many places, the article is written chaotically, i.e. missing punctuation and units (line 129, etc.).
4. Are you express Ki-67 (Table 1) in this way or Ki67 (Table 2)?
5. The discussion should definitely be expanded. Using only 5 references is definitely not enough.
6. Also, in discussion section please add paragraph related to clinical and practical aspects of the study.
7. How we can applicate your results into practice?, why your work is valuable in the field? Please provide separate paragraph in this regard.
8.Conclusions should be provided in separate paragraph and the authors should clearly present a summary of the research.

Minor points:
1.  Modification of the grammar and punctuation is required.
2.  Please provide strengths and limitations of the study.
3. The article was prepared contrary to the applicable guide for authors.

General: interesting, but chaotic work.

Comments on the Quality of English Language

Modification of the grammar and punctuation is required.

Author Response

Rebuttal letter

REVIEWER 1

We thank the reviewer for the positive comment.

We have provided a point-by-point response.

We hope that now the manuscript can be suitable for publication.

Major:
1. In Methods section, authors should provide subsections, including study design, inclusion and exclusion criteria, etc.

Reply: It has been done. We have added the study design and the inclusion and exclusion criteria in the Materials and Methods section on page 2 and 3, lines 78-94.
2. In my version of the pdf file, figure 1 is incomplete. Figure 2 is also incomplete.

Reply: They have been modified.

  1. In many places, the article is written chaotically, i.e. missing punctuation and units (line 129, etc.).

Reply: We have corrected the errors and rewritten the phrase in line 129.
4. Are you express Ki-67 (Table 1) in this way or Ki67 (Table 2)?

Reply: We expressed Ki67 the same way throughout the text.
5. The discussion should definitely be expanded. Using only 5 references is definitely not enough.

Reply: We have fully expanded the discussion and added more references (n 26,27,28,29, and 31).
6. Also, in discussion section please add paragraph related to clinical and practical aspects of the study.

Reply: We have added a paragraph (n 5.1) on the clinical and practical aspects of the study on page 9, lines 222-231.
7. How we can applicate your results into practice?, why your work is valuable in the field? Please provide separate paragraph in this regard.

Reply:  The strength of this study has been outlined on page 8 and 9, lines 213-219, and information on how our results could be applied to clinical practice can be also found in the conclusion on page 9, lines 222-231.
8.Conclusions should be provided in separate paragraph and the authors should clearly present a summary of the research.

Reply: Conclusions have been provided in a separate paragraph on page 9, lines 232-239.

Minor points:
1.  Modification of the grammar and punctuation is required.

Reply: The manuscript has been fully revised by a native speaker.
2.  Please provide strengths and limitations of the study.

The strength of this study has been outlined on page 8 and 9, lines 213-219, and the limitations at page 9 lines 211-212.

  1. The article was prepared contrary to the applicable guide for authors.

Reply: We have since followed the guidelines for authors.

Reviewer 2 Report

Comments and Suggestions for Authors

1. In the first line of Introduction please mentioned the time period.

2. The results section is not organized, author may give heading in the result sections. 

3. I have did not find conclusion section in after discussion. 

4. There are frequent typo error throughout the manuscript. 

Author Response

REVIEWER 2

Rebuttal letter

We have provided a point-by-point response.

We hope that now our manuscript can be suitable for publication.

Comments and Suggestions for Authors

  1. In the first line of Introduction please mentioned the time period.

Reply: It has been done. We added the year 2023 in the first line of the introduction.

  1. The results section is not organized, author may give heading in the result sections. 

Reply: It has been done.

  1. I have did not find conclusion section in after discussion. 

Reply: The conclusion section has been added after the discussion.

  1. There are frequent typo error throughout the manuscript. 

Reply: The typos have been corrected by the authors and by a native English speaker.

Reviewer 3 Report

Comments and Suggestions for Authors

The aim of this retrospective case series study was to identify clinicopathological characteristics that may be useful in predicting the response to CDK4/6 inhibitors among patients with hormone receptor positive (HR+) metastatic breast cancer (mBC). The study included 177 mBC patients, 66 of whom were treated with CDK4/6 inhibitors plus letrozole and 111 who were treated with CDK4/6 inhibitors and fulvestrant. A multistate model was used to analyze the data. The results revealed that low body surface area and older age were significantly associated with an increased risk of developing neutropenia, a common side effect of CDK4/6 inhibitors. Additionally, high levels of Ki67, the presence of visceral metastases, and the absence of prior adjuvant chemotherapy were identified as prognostic factors for disease progression or death. Furthermore, among patients who developed neutropenia, those with multiple previous lines of treatment had a higher risk of disease progression or death. Importantly, patients with neutropenia had more than twice the risk of disease progression or death compared to those without neutropenia. Based on these findings, the authors suggest that healthcare professionals should consider these baseline characteristics when treating patients with HR+ mBC, as they could potentially reduce the occurrence of both neutropenia and disease progression. Further research is needed to validate these findings and determine their applicability in clinical practice. This is an interesting and infromative manuscript. But I have several following concerns:

1. Abbreviations should be defined when they first appear in the text. Such as "CDK4/6", "mTOR",...

2. In Line 4, you'd better remove the ":" and the "." at the end of the sentence.

3. Represent significant difference of "P" should be italic.

4. Records modified with delete lines should be removed in the manuscript to avoid confusion.

5. The information of two pictures in the text is partially obscured. Please show all the picture content in the text and improve the resolution of the picture. In addition, the full text may have accidentally changed the format when using the submission template. The format of the whole article is a little strange, please check and modify it carefully.

6. The legend of each figure should follow the corresponding Figure.

7. Tables should use a standard three-line table and should not span pages.

8. The nucleic acid sequences (including gene names, regulatory sequences, and primer names) should be in italics. 

9.  Please unify the format of references in the article, including the author's name, the case of words in the title of the article, the writing of the name of the journal, and the page number.

Comments on the Quality of English Language

Moderate editing of English language required

Author Response

REVIEWER 3

Rebuttal letter

We have provided a point-by-point response

We hope that now our manuscript can be suitable for publication.

  1. Abbreviations should be defined when they first appear in the text. Such as "CDK4/6", "mTOR",.

Reply: It has been done.

  1. In Line 4, you'd better remove the ":" and the "." at the end of the sentence.

Reply: They have been removed.

  1. Represent significant difference of "P" should be italic.

Reply: Now p is in italic.

  1. Records modified with delete lines should be removed in the manuscript to avoid confusion.

Reply: It has been done.

  1. The information of two pictures in the text is partially obscured. Please show all the picture content in the text and improve the resolution of the picture. In addition, the full text may have accidentally changed the format when using the submission template. The format of the whole article is a little strange, please check and modify it carefully.

Reply: It has been done.

  1. The legend of each figure should follow the corresponding Figure.

Reply: It has been done.

  1. Tables should use a standard three-line table and should not span pages.

Reply: It has been done.

  1. The nucleic acid sequences (including gene names, regulatory sequences, and primer names) should be in italics.

Reply: It has been done.

  1. Please unify the format of references in the article, including the author's name, the case of words in the title of the article, the writing of the name of the journal, and the page number.

Reply: It has been done.

Round 2

Reviewer 1 Report

Comments and Suggestions for Authors

The authors have revised the article in accordance with the suggestions, and on this basis I suggest publication in its current form.

Reviewer 2 Report

Comments and Suggestions for Authors

Author(s) revised the manuscript as commented. 

Reviewer 3 Report

Comments and Suggestions for Authors

The authors have addressed all my concerns. I recommend accepting this manuscript in present form.